

# Level of anxiety among healthcare providers during COVID-19 pandemic in Saudi Arabia: cross-sectional study

Abbas Al Mutair[1,2,3], Alya Al Mutairi[4], Yasmine Alabbasi[5], Abbas Shamsan[6], Sana Al-Mahmoud[7], Saad Alhumaid[8], Muhammad zeshan Arshad[9], Mansour Awad[10] and Ali Rabaan[11,12]

[1] College of Nursing, Princess Norah University, Riyadh, Saudi Arabia
[2] School of Nursing, University of Wollongong, Wollongong, Australia
[3] Research Center, Almoosa Specialist Hospital, Al-ahsa, Saudi Arabia, Al-Ahsa, Saudi Arabia
[4] Department of Mathematics, Faculty of Science, Taibah University, Medina, Saudi Arabia
[5] Maternity and Child Health Nursing Department, College of Nursing, Princess Nourah bint Abdulrahman University, Riyadh, Saudi Arabia
[6] Research Center, Dr. Sulaiman Al Habib Medical Group, Riyadh, Saudi Arabia
[7] Imam Abdurrahman Bin Faisal University, Riyadh, Saudi Arabia
[8] Administration of Pharmaceutical Care, Al-Ahsa Health Cluster, Ministry of Health, Riyadh, Saudi Arabia
[9] Department of Mathematics and Statistics, University of Agriculture, Faisalabad, Pakistan
[10] Commitment Administration, General Directorate of Health Affairs, Medina, Medina, Ministry of Health, Saudi Arabia
[11] Molecular Diagnostic Laboratory, Johns Hopkins Aramco Healthcare, Dhahran, Saudi Arabia
[12] Department of Public Health and Nutrition, the University of Haripur, Haripur, Pakistan

Corresponding author
Alya Al Mutairi,
amutairi@taibahu.edu.sa

## ABSTRACT

**Background:** The burden of the spread of the COVID-19 pandemic has impacted widely on the healthcare providers physically and mentally. Many healthcare providers are exposed to psychological stressors due to their high risk of contracting the virus.

**Aims:** This study aimed to measure the level of anxiety among healthcare providers during the COVID-19 pandemic in Saudi Arabia. In addition, this study aimed to measure the level of anxiety based on demographic characteristics.

**Method:** A cross-sectional survey was employed to recruit a convenience sample of healthcare providers. A pencil and paper self-administered questionnaires were used to collect data from demographic and generalized anxiety disorder GAD-7 data. However, this study received written informed consent from participants of the study. In addition, the study was approved by the Institutional Review Board at Dr. Sulaiman Al Habib Medical Group (IRB Log No. RC20.06.88-03).

**Results:** A total of 650 participants were recruited, results of GAD-7 showed that 43.5%, 28.9% and 27.5% of healthcare providers in Saudi Arabia experienced mild, moderate and severe anxiety, respectively, during the COVID-19 pandemic. Results indicated that age, health specialty, nationality, and sleeping disorders before COVID-19 were associated with anxiety levels.

**Conclusion:** The generalized anxiety among healthcare providers in Saudi Arabia was mild. Older healthcare providers were found to have a higher level of anxiety compared to other participating healthcare providers. Several factors may contribute to a higher level of anxiety including age, socioeconomic status, marital status, having chronic conditions, and sleeping disorder before the COVID-19 pandemic.

> To further understand the level of anxiety among healthcare providers during the COVID-19 pandemic in Saudi Arabia, longitudinal and mixed-method research is needed.

## BACKGROUND

The coronavirus disease of 2019 (COVID-19) outbreak has spread across the world; besides, there is a degree of uncertainty, concern, and worry among healthcare providers (*World Health Organization, 2021*; *Mental Health America, 2020*). These excessive worries, intrusive thoughts, and stress may have an impact on the level of anxiety and mental health of healthcare providers (*Mental Health America, 2020*; *American Psychological Association, 2021*). In times of pandemics, such as COVID-19, healthcare providers, as the frontline force, may be prone to mental stress due to uncertainty about infectious disease and fear of contracting the virus and transmitting it to loved ones (*Wang et al., 2020*). Workplace stress in healthcare industry is persistent due to several reasons including exposing to infectious diseases leading to illness or death (*Al Mutair et al., 2021*). This may produce high rates of anxiety and depression among health care providers (*Magnavita et al., 2021*). Burnout among health care providers is also dominant due to infection and other stressors such as shortage of staff, patients load or long working hours (*Al Mutair et al., 2021*). There is growing evidence of that pandemic such as COVID-19 produces high burnout level among healthcare providers and may impact negatively on the healthcare workers mental health and emotional wellbeing (*Chirico & Magnavita, 2020*; *Chirico et al., 2021*; *Chirico & Nucera, 2020*).

A national survey evaluating the psychological impact of COVID-19 among the general public in China during the initial stages of the COVID-19 outbreak found that 28% of respondents reported moderate to severe anxiety symptoms (*Wang et al., 2020*). Another study in China found that the prevalence of anxiety was 44.7% (GAD 7 ≥ 5) among healthcare providers (*Zhang & Ma, 2020*). A survey from Mental Health America (MHA) reported that 86% of healthcare providers regularly experienced anxiety from June 2020 to September 2020 during the COVID-19 outbreak (*Mental Health America, 2020*). In Saudi Arabia, the first case of COVID-19 infection was reported on March 02, 2020, amid growing concerns and uncertainties among the community and healthcare providers (*Ministery of Health, 2020*). COVID-19 cases which were reported in Saudi Arabia vary in their severities from mild, moderate to severe (*Al Mutair et al., 2020b*; *Al-Omari et al., 2020b*). Healthcare providers in Saudi Arabia are exposed to a high level of stressors due to high susceptibility to getting infected which may result in a high level of burnout (*Al Mutair et al., 2018*; *Al Mutair et al., 2020a*; *Al-Omari et al., 2020a*; *Al Mutair et al., 2017*; *Fernandez et al., 2020*). Most studies have assessed the effect of COVID-19 on the anxiety and mental health of healthcare providers in China (*De Kock et al., 2021*).

However, to our knowledge, there are limited studies evaluating anxiety among healthcare providers during the COVID-19 pandemic in Saudi Arabia. Therefore, the purpose of this study is to measure the level of anxiety among healthcare providers during the COVID-19 pandemic in Saudi Arabia. In addition, this study aims to measure the level of anxiety based on demographic characteristics. We hypothesized that levels of anxiety among healthcare providers were associated with their demographic characteristics during the COVID-19 pandemic in Saudi Arabia.

### Aim of the study

To measure the level of anxiety among healthcare providers during the COVID-19 pandemic in Saudi Arabia. Besides that, this study aimed to measure the level of anxiety based on demographic characteristics.

## METHODS

### Study design

A cross-sectional survey study was employed to recruit a convenience sample of healthcare providers during the COVID-19 pandemic in Saudi Arabia. Prior to data collection, and ethical approval to conduct the study was sought from the Institutional Review Board at Dr. Sulaiman Al Habib Medical Group (IRB Log No. RC20.06.88-03). In addition, participants were ensured that taking part in the study is voluntary and that all gathered information will only be used for the study purposes and will be kept secured and confidential. For this cross-sectional study, no informed consent was deemed necessary and was waived by the IRB. Data were identified for the use of this publication and the study adhered to the ethical guidelines of the Declaration of Helsinki and good clinical practice.

Both Saudi and non-Saudi healthcare providers were invited to participate in the current study. Participants were included if they met the following criteria: 22 years old or above, responsible for providing direct patient care in an inpatient or outpatient healthcare setting, and spent at least six months in the current clinical unit. The sample size was estimated using G*Power3 and based on the confidence level of 95%, power of 80%, and medium effect size as determined by the literature review. The minimum required sample size was 356 subjects. A total of 900 questionnaires were distributed among healthcare providers who work in the private and public healthcare sector in Riyadh city in Saudi Arabia between April 1 and 15, 2020. A total of 650 participants returned the completed surveys giving a response rate of 72% (Fig. 1).

### Data collection instrument

A pencil and paper self-administered questionnaire were used to collect data from the participants. The questionnaire consisted of socio-demographic characteristics including: age, gender, nationality, working area, profession, type of healthcare facility and years of working experience. The questionnaire also consisted of Generalized Anxiety Disorder GAD-7. The Generalized Anxiety Disorder Scale-7 CAD-7 was developed by Spitzer and colleagues (*Spitzer et al., 2006*). It is self-rated scales and consists of 7 items which have

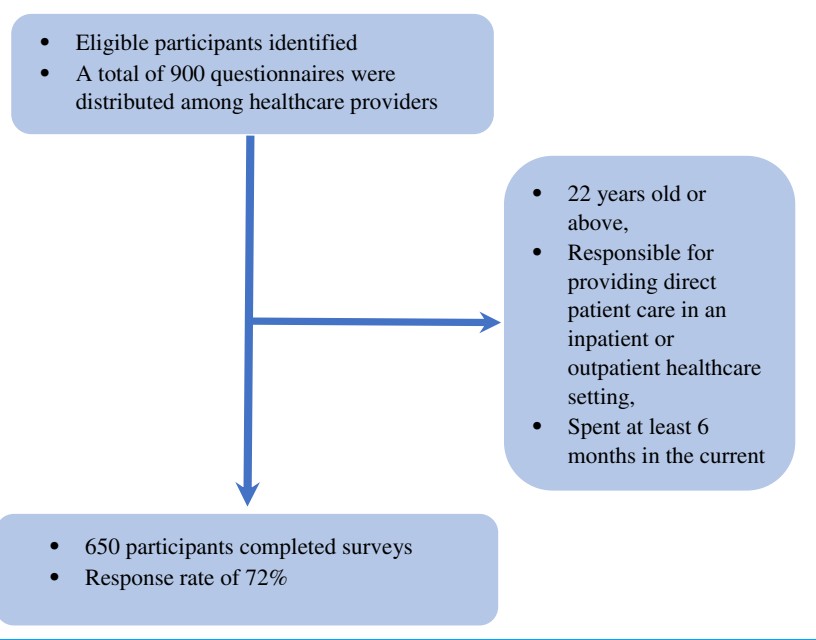

**Figure 1 Flow chart.**

been used in multiple studies and had shown acceptable reliability and good validity for assessing anxiety disorder in clinical practice and research (*Rutter & Brown, 2017*). CAD-7 items are rated on a 4-point Likert Scale (0 = not at all, 1 = several days, 2 = over half the days and 3 = nearly every day). The scale items describe the diagnostic features of the generalized anxiety disorder. The scale scores can range from 0 indicating no anxiety symptoms to 21 indicating more sever anxiety symptoms.

## Data analysis

In this study standard statistical procedures were applied, data was collated into an excel spread sheet and then imported into the Statistical Package for Social Sciences (SPSS, version 25). The data was validated for accuracy and completeness before conducting the statistical analysis. A detailed descriptive and inferential statistics for all the variables constituting the socio-demographic characteristics questionnaire and Generalized Anxiety Disorder GAD-7. A descriptive analysis for socio-demographic and perceptional variables was completed in order to capture frequencies, means and standard deviations. Frequency distributions for continuous variables were examined *via* Shapiro-Wilk test and appropriate statistical tests were applied accordingly. An inferential statistics chi-square analysis was employed to study the association between demographic profile and GAD-7. $P$-values of $\leq 0.05$ were accepted as the significance level for all inferential statistical tests that were conducted.

## RESULTS

Demographic profile has been analyzed and the findings are presented in Table 1. Several demographic profiles have been chosen, namely type of health care facility, age, gender, nationality, the health specialty, hospital department, experiences (years), sleeping

**Table 1 Demographic characteristics of the respondents.**

| Demography profile | General anxiety disorder (GAD-7) | | | | |
| --- | --- | --- | --- | --- | --- |
| | Mild | Moderate | Severe | *n* | % |
| **Type of facility** | | | | | |
| Government | 86 (42.6%) | 66 (32.7%) | 50 (24.8%) | 202 | 31.3% |
| Private | 194 (43.7%) | 121 (27.3%) | 129 (29.1%) | 444 | 68.7% |
| **Age** | | | | | |
| 20–30 years old | 70 (56.9%) | 32 (26%) | 21 (17.1%) | 123 | 32.4% |
| 31–40 years old | 73 (41.2%) | 56 (31.6%) | 48 (27.1%) | 177 | 46.6% |
| 41–50 years old | 34 (54%) | 14 (22.2%) | 15 (23.8%) | 63 | 16.6% |
| >50 years old | 12 (70.6%) | 4 (23.5%) | 1 (5.9%) | 17 | 4.5% |
| **Gender** | | | | | |
| Male | 78 (46.7%) | 52 (31.1%) | 37 (22.2%) | 167 | 26.0% |
| Female | 199 (41.9%) | 134 (28.2%) | 142 (29.9%) | 475 | 74.0% |
| **Nationality** | | | | | |
| Saudi | 66 (34.9%) | 67 (35.4%) | 56 (29.6%) | 189 | 29.6% |
| Non Saudi | 210 (46.7%) | 119 (26.4%) | 121 (26.9%) | 450 | 70.4% |
| **Profession** | | | | | |
| Physician | 45 (39.5%) | 37 (32.5%) | 32 (28.1%) | 114 | 17.7% |
| Nurse | 159 (49.2%) | 81 (25.1%) | 83 (25.7%) | 323 | 50.2% |
| Others | 76 (36.7%) | 70 (33.8%) | 61 (29.5%) | 207 | 32.1% |
| **Work area** | | | | | |
| ER | 18 (34.6%) | 21 (40.4%) | 13 (25%) | 52 | 8.2% |
| Ward | 71 (53%) | 28 (20.9%) | 35 (26.1%) | 134 | 21.1% |
| ICU | 100 (41.2%) | 72 (29.6%) | 71 (29.2%) | 243 | 38.3% |
| Others | 86 (41.7%) | 63 (30.6%) | 57 (27.7%) | 206 | 32.4% |
| **Years of experience** | | | | | |
| 1–5 years | 126 (45%) | 85 (30.4%) | 69 (24.6%) | 280 | 44.3% |
| 6–10 years | 64 (36.2%) | 54 (30.5%) | 59 (33.3%) | 177 | 28.0% |
| 11 years and above | 84 (48%) | 46 (26.3%) | 45 (25.7%) | 175 | 27.7% |
| **Sleeping disorder*** | | | | | |
| Yes | 21 (26.3%) | 32 (40%) | 27 (33.8%) | 80 | 13.1% |
| No | 250 (47%) | 153 (28.8%) | 129 (24.2%) | 532 | 86.9% |
| **Mental disorder*** | | | | | |
| Yes | 4 (20%) | 7 (35%) | 9 (45%) | 20 | 3.3% |
| No | 268 (45.3%) | 175 (29.6%) | 149 (25.2%) | 592 | 96.7% |

Note:
* Before covid-19 incident.

disorder before COVID-19 and mental disorder. In terms of healthcare facilities, 444 (68.7%) were private facilities while only 202 (31.3%) government. By looking at the age, about 177 (46.6%) of the respondents were at the age range 31–40 years old, followed by 123 (32.4%) were 20–30 years old, 63 (16.6%) were 41–50 years old and small percentage 17 (4.5%) were above 50 years old. It was reported that more females 475 (74%)

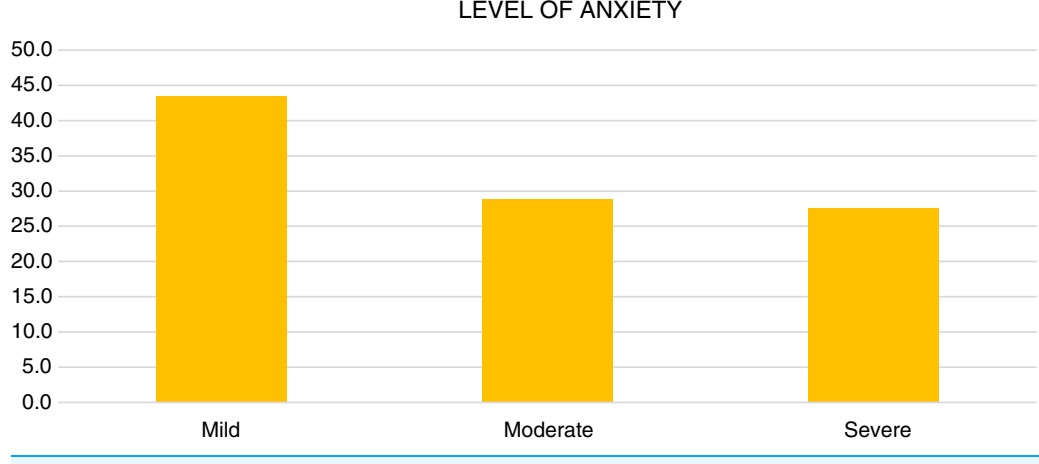

**Figure 2 Level of anxiety.**

**Table 2 Level of anxiety.**

| Level of anxiety | Frequency | Percent |
| --- | --- | --- |
| Mild | 283 | 43.5 |
| Moderate | 188 | 28.9 |
| Severe | 179 | 27.5 |
| Total | 650 | 100.0 |

than males 167 (26%). Almost three-quarters of the respondents 450 (70.4%) were non-Saudi while 189 (26%) were Saudi. By studying the health specialty, about 323 (50.2%) of the respondents were nurses, followed by 207 (32.1%) who were others and 114 (17.7%) physicians. Approximately 243 (38.3%) of the respondents worked at ICU department, followed by 206 (32.4%) worked in other departments, 134 (21.1%) wards and 52 (8.2%) worked at Emergency department. By studying the year of experience, a total of 280 (44.3%) with working experience 1–5 years, followed by 177 (28%) worked 6-10 years and 175 (27.7%) worked 11 years and above. Respondents were asked about sleeping disorder before COVID-19, about 532 (86.9%) did not have sleeping disorder while 80 (13.1%) had sleeping disorder before COVID-19. Lastly, in terms of mental disorder, about 592 (96.7%) did not report any mental disorder, while 20 (3.3%) had mental disorder.

## Level of anxiety

Level of anxiety has been divided into three levels: mild, moderate and sever anxiety. The findings demonstrated about 283 (43.5%) had mild anxiety, followed by 188 (28.9%) had moderate anxiety and 179 (27.5%) had severe anxiety as shown in Table 2 and Fig. 2.

## The association between demographic characteristics and level of anxiety

Chi-square analysis has been computed to examine the association between level of anxiety and demographic profile (see Table 3). As reported, age ($X2 = 12.892$, df = 6, $p < 0.05$), nationality ($X2 = 8.321$, df = 2, $p < 0.05$), health specialty ($X2 = 9.543$, df = 4, $p < 0.05$) and

**Table 3 The association between demographic characteristics and level of anxiety.**

| Variables | Mild | | Moderate | | Severe | | Value | df | sig. |
|---|---|---|---|---|---|---|---|---|---|
| | n | % | N | % | N | % | | | |
| **Type of health care facilities** | | | | | | | 2.377 | 2 | 0.305 |
| Government | 86 | 42.6 | 66 | 32.7 | 50 | 24.8 | | | |
| Private | 194 | 43.7 | 121 | 27.3 | 129 | 29.1 | | | |
| **Age** | | | | | | | 12.892 | 6 | 0.045 |
| 20–30 | 70 | 56.9 | 32 | 26.0 | 21 | 17.1 | | | |
| 31–40 | 73 | 41.2 | 56 | 31.6 | 48 | 27.1 | | | |
| 41–50 | 34 | 53.97 | 14 | 22.2 | 15 | 23.8 | | | |
| Above 50 years | 12 | 70.6 | 4 | 23.5 | 1 | 5.9 | | | |
| **Gender** | | | | | | | 3.683 | 2 | 0.159 |
| Male | 78 | 46.7 | 52 | 31.1 | 37 | 22.2 | | | |
| Female | 199 | 41.9 | 134 | 28.2 | 142 | 29.9 | | | |
| **Nationality** | | | | | | | 8.321 | 2 | 0.016 |
| Saudi | 66 | 34.9 | 67 | 35.4 | 56 | 29.6 | | | |
| Non-Saudi | 210 | 46.7 | 119 | 26.4 | 121 | 26.9 | | | |
| **Health specialty** | | | | | | | 9.543 | 4 | 0.049 |
| Physicians | 45 | 39.5 | 37 | 32.5 | 32 | 28.1 | | | |
| Nurses | 159 | 49.2 | 81 | 25.1 | 83 | 25.7 | | | |
| Others | 76 | 36.7 | 70 | 33.8 | 61 | 29.5 | | | |
| **Hospital Department** | | | | | | | 10.216 | 6 | 0.116 |
| ER Department | 18 | 34.6 | 21 | 40.4 | 13 | 25.0 | | | |
| Ward | 71 | 53.0 | 28 | 20.9 | 35 | 26.1 | | | |
| ICU Department | 100 | 41.2 | 72 | 29.6 | 71 | 29.2 | | | |
| Others | 86 | 41.7 | 63 | 30.6 | 57 | 27.7 | | | |
| **Years of experience** | | | | | | | 7.134 | 4 | 0.129 |
| 1–5 years | 126 | 45.0 | 85 | 30.4 | 69 | 24.6 | | | |
| 6–10 years | 64 | 36.2 | 54 | 30.5 | 59 | 33.3 | | | |
| 11 years and above | 84 | 48.0 | 46 | 26.3 | 45 | 25.7 | | | |
| **Sleeping disorder before Covid-19** | | | | | | | 12.127 | 2 | 0.002 |
| Yes | 21 | 26.3 | 32 | 40.0 | 27 | 33.8 | | | |
| No | 250 | 47.0 | 153 | 28.8 | 129 | 24.2 | | | |
| **Mental Disorder** | | | | | | | 5.919 | 2 | 0.052 |
| Yes | 4 | 20.0 | 7 | 35.0 | 9 | 45.0 | | | |
| No | 268 | 45.3 | 175 | 29.6 | 149 | 25.2 | | | |

sleeping disorder before COVID-19 (X2 = 12.127, df = 2, $p < 0.05$) have significant association with anxiety level. While type of health care facilities (X2 = 2.377, df = 2, $p > 0.05$), gender (X2 = 3.683, df= 2, $p > 0.05$), hospital department (X2 = 10.216, df = 6, $p > 0.05$), years of experience (X2 = 7.134, df = 4, $p > 0.05$) and mental disorder (X2 = 5.919, df = 2, $p > 0.05$) have no significant association with level of anxiety. Descriptive analysis is shown in Table 4. In terms of age, we found that respondents at the

| Table 4 Descriptive analysis (frequency, percentage, mean and SD). | | | | |
|---|---|---|---|---|
| Demographic profile | N | % | Mean | SD |
| **Type of health care facility** | | | | |
| Government | 202 | 31.3 | 7.39 | 5.34 |
| Private | 444 | 68.7 | 7.73 | 6.02 |
| **Age** | | | | |
| 20–30 years old | 123 | 32.4 | 6.24 | 4.92 |
| 31–40 years old | 177 | 46.6 | 7.62 | 5.37 |
| 41–50 years old | 63 | 16.6 | 5.98 | 5.18 |
| Above 50 years old | 17 | 4.5 | 4.29 | 4.00 |
| **Gender** | | | | |
| Male | 167 | 26.0 | 6.79 | 5.61 |
| Female | 475 | 74.0 | 7.96 | 5.86 |
| **Nationality** | | | | |
| Saudi | 189 | 29.6 | 8.33 | 5.54 |
| Non Saudi | 450 | 70.4 | 7.34 | 5.88 |
| **Health specialty** | | | | |
| Physicians | 114 | 17.7 | 7.66 | 5.70 |
| Nurses | 323 | 50.2 | 7.26 | 5.68 |
| Others | 207 | 32.1 | 8.05 | 6.02 |
| **Hospital Department** | | | | |
| ER Department | 52 | 8.2 | 7.73 | 5.65 |
| Ward | 134 | 21.1 | 7.21 | 6.05 |
| ICU Department | 243 | 38.3 | 7.75 | 5.53 |
| Others | 206 | 32.4 | 7.72 | 6.01 |
| **Years of experience** | | | | |
| 1–5 years | 280 | 44.3 | 7.43 | 5.62 |
| 6–10 years | 177 | 28.0 | 8.68 | 6.16 |
| 11 years and above | 175 | 27.7 | 6.78 | 5.59 |
| **Sleeping disorder before covid-19** | | | | |
| Yes | 80 | 13.1 | 8.98 | 5.33 |
| No | 532 | 86.9 | 7.05 | 5.51 |
| **Mental Disorder** | | | | |
| Yes | 20 | 3.3 | 11.11 | 6.39 |
| No | 592 | 96.7 | 7.20 | 5.49 |

age 31–40 years (46.6 ± 7.62) perceived anxiety significantly higher than above 50 years old (4.5 ± 4.29). Additionally, Saudi perceived more anxiety (8.33 ± 5.54) as compared to non-Saudi (7.33 ± 5.88). In terms of health specialty, others healthcare providers (8.05 ± 6.01) exhibited higher anxiety level as compared to nurses (7.26 ± 5.68). Also, those who had sleeping disorders before COVID-19 (8.98 ± 5.32) perceived higher anxiety than those who did not have sleeping disorder (7.04 ± 5.51).

**Table 5 Model summary statistic.**

| Model summary | Chi-square | df | p Value | R2 |
|---|---|---|---|---|
| **Model fitting information** | | | | |
| Final model | 19.292 | 3 | <0.001 | |
| **Goodness-of-Fit** | | | | |
| Pearson | 10.897 | 11 | 0.452 | |
| Deviance | 11.231 | 11 | 0.424 | |
| **Pseudo R-square** | | | | |
| Cox and Snell | | | | 0.031 |
| Nagelkerke | | | | 0.035 |
| McFadden | | | | 0.015 |
| **Test of parallel lines** | | | | |
| General | 5.826 | 3 | 0.120 | |

Note:
Link function: Logit.

**Table 6 Summary statistic for ordinal logistic regression estimation.**

| Parameter estimates | B | OR | 95% CI | | Wald | df | p Value |
|---|---|---|---|---|---|---|---|
| | | | Lower | Upper | | | |
| **Threshold** | | | | | | | |
| (LEVEL_ANXIETY = 1.00) | −0.121 | | | | 1.4 | 1 | 0.245 |
| (LEVEL_ANXIETY = 2.00) | 1.221 | | | | 110.7 | 1 | 0.000 |
| **Location** | | | | | | | |
| (Gender = 1) Male | −0.388 | 0.678 | 0.475 | 0.970 | 4.5 | 1 | 0.034 |
| (Gender = 2) Female (ref.) | | | | | | | |
| (Nationality = 1) Saudi | 0.460 | 1.584 | 1.127 | 2.226 | 7.0 | 1 | 0.008 |
| (Nationality = 2) Non Saudi (ref.) | | | | | | | |
| (SleepingDisorder = 1) Yes | 0.555 | 1.742 | 1.119 | 2.713 | 6.0 | 1 | 0.014 |
| (SleepingDisorder = 2) No (ref.) | | | | | | | |

Note:
Link function: Logit.

## Effect of demographic factors on level of anxiety

The Chi-square test showed that working experience and age groups were highly associate ($p < 0.001$). The results showed that older age groups were associate with longer working experience. The final results of ordinal logistic regression were summarized and presented in Tables 5 and 6. In the final model, only three significant factors were remained; gender, nationality, and sleeping disorder. They were found to significantly explain the odds of having general anxiety disorder. Based on summary statistic, final model was significant ($X2 = 19.292$, df = 3, $p < 0.05$). As for goodness of fit index, both Pearson ($X2 = 10.897$, df = 11, $p > 0.05$) and Deviance ($X2 = 11.231$, df = 11, $p > 0.05$) showed that the final model has good fit with the data. Lastly, the assumption of proportional odds (parallel lines test) showed that the assumption was met ($X2 = 5.826$, df = 3, $p > 0.05$). Therefore, the model results can be used for interpretation. Gender has

effect on participants' anxiety level, the *P* value of −0.38 indicating that male participants are less likely to classify with higher level of anxiety compared to female (Table 6). The results showed that that the odds of male participants to have higher anxiety level are 0.67 (95% CI [0.475–0.970]) times than that of female respondents with significant statistic values of Wald $\chi^2$ (1) = 4.5, *p* = 0.034. Additionally, Saudi participants are more likely to develop higher level of anxiety compared to non-Saudi 1.584 (95% CI [1.127–2.226]) times higher than non-Saudi respondents, supported with statistical values of Wald $\chi^2$ (1) = 7.0, *p* = 0.008. The results showed that participants with sleeping disorder before COVID-19 are prone to classify with higher level of anxiety compared to those who do not have such disorder 1.742 (95% CI [1.119–2.713]) times higher than those without sleeping disorder, with statistical significance value of Wald $\chi^2$ (1) = 6.0, *p* = 0.014.

## DISCUSSION

During the novel coronavirus pandemic healthcare providers are risking their lives and continue working with tremendous efforts towards their ethical and professional obligations. Not only do healthcare providers be under psychological distress during the epidemic, but also psychological consequences might accrue on the long-term (*Al Mutair et al., 2021*; *Al Mutair et al., 2020b*). This study measured the level of anxiety based on demographic characteristics among healthcare providers during the COVID-19 pandemic in Saudi Arabia. Survey results showed that 43.5%, 28.9% and 27.5% of healthcare providers in Saudi Arabia experienced mild, moderate, and severe anxiety, respectively, during the pandemic. Even though, during MERS-CoV epidemic in Saudi Arabia, research show that hospital staff faced stressful times, and they felt anxious, nervous, and emotionally distressed (*Al Mutair & Ambani, 2020*). Also, Unlike SARS and Ebola *versus* there were psychological consequences among hospital staffs (*Lin et al., 2007*; *Lehmann et al., 2015*). However, same finding were found in a recent meta-analysis which showed that lower rates of anxiety and depression during COVID–19 than the reported rates among healthcare providers during and after MERS and SARS (*Pappa et al., 2020*). There were associations between some demographic characteristics and the level of anxiety. Age, health specialty, nationality, and sleeping disorders before COVID-19 were associated with anxiety levels, whereas other demographic characteristics, such as type of healthcare facility, gender, hospital department, years of experience, and mental disorders, did not influence anxiety levels. This study found that respondents between the ages of 31–40 years experienced significantly higher anxiety than respondents above 50 years old. A comparable study conducted in Saudi Arabia in March 2020 measured depression and anxiety among healthcare providers and found that participants between 30 and 39 years old were significantly associated with anxiety (7.40 ± 6.59, *p* < 0.001) (*Al-Omari et al., 2020a*). Contrary to previous studies (*Al Mutair et al., 2017*; *Fernandez et al., 2020*; *De Kock et al., 2021*) our study found that healthcare providers other than nurses showed higher anxiety levels (8.05 ± 6.01) compared to nurses (7.26 ± 5.68). A possible reason was the difference in demographic characteristics. The study sample mainly consisted of non-Saudi healthcare providers and many

non-Saudi healthcare providers may live away from their loved ones. This may contribute to anxiety and depression level increase among healthcare providers, as non-Saudi mainly live alone and away from their family which may result in high level of anxiety and psychological distress (*Al Mutair et al., 2020b*). Our study further measured the level of anxiety based on nationality. Findings showed that anxiety levels among Saudi healthcare providers ($8.33 \pm 5.54$) were significantly higher than non-Saudi providers ($7.33 \pm 5.88$). Many factors may contribute to higher anxiety levels among Saudi healthcare providers, regardless of their professional designation. These include socioeconomic status, marital status, having a chronic health condition, fear of contracting the virus, living with an immunocompromised, chronically ill, or elderly person (*AlAteeq et al., 2020*; *Lai et al., 2020*; *Alenazi et al., 2020*). A Saudi-based study measuring the overall emotional wellbeing and its predictors of the Saudi population during the COVID-19 pandemic found that age, gender, marital status, and socioeconomic status are majors' predictors of emotional wellbeing (*Al Mutair, Alhajji & Shamsan, 2021*). Similar concerns and fears about transmitting the virus to their families were reported among healthcare providers during the severe acute respiratory syndrome (SARS) outbreak in 2003 (*Al Mutair, Alhajji & Shamsan, 2021*). Cultural norms and differences in living conditions among Saudi and non-Saudi healthcare providers may contribute to higher anxiety levels. In terms of sleeping disorders, this study found that healthcare providers who had sleeping disorders before COVID-19 exhibited higher anxiety ($8.98 \pm 5.32$) compared to those who did not have sleeping disorders ($7.04 \pm 5.51$) ($X2 = 12.127$, $df = 2$, $p < 0.05$). This is important because a systematic review and meta-analysis found that the prevalence of sleep disturbances during COVID-19 was approximately 34.8% among nurses in six of the reviewed studies and 41.6% among physicians in four of the studies reviewed (*Maunder et al., 2003*). Previous studies found that sleeping disorders among healthcare providers were associated with an 83% event of adverse safety outcomes, such as motor vehicle crashes, exposure to potentially infectious materials, and medical errors (*Salari et al., 2020*; *Weaver et al., 2018*). Moreover, screening positive for anxiety or depression increased the risk of adverse safety outcomes by 63% (*Al Mutair et al., 2019*; *Bové et al., 2014*). Anxiety among healthcare providers accompanied by sleeping disorders during the pandemic crisis may interfere with physical, mental, and emotional functioning, and result in adverse occupational safety outcomes. Possible limitations of this study include its cross-sectional design, as casual inferences should not be made. The limitations also relate to the GAD-7 scale, as it screens for an anxiety disorder and provides a probable diagnosis that must be confirmed with a physical examination or blood tests to rule out thyroid dysfunction (*Al Mutair et al., 2020c*; *Brandt et al., 2014*). Since recruitments were made *via* questionnaire, healthcare providers who responded may be more self-aware and interested in revealing their concerns. This study represented Saudi and non-Saudi healthcare providers, so there may be cultural norms and differences in living conditions that might have affected our findings. Future research may use a longitudinal study design to understand the pattern of the levels of anxiety among healthcare providers over time during the COVID-19 pandemic. In addition, further research may use a mixed-method

design that allows qualitative and quantitative data integration, thus providing a broader, in-depth knowledge of the effect of COVID-19 on anxiety levels among healthcare providers. Future studies may also compare the effects of COVID-19 on anxiety levels among Saudi and non-Saudi healthcare providers.

## CONCLUSION

In conclusion, this Saudi-based study identified that healthcare providers' overall generalized anxiety disorder during the COVID-19 pandemic was classified as mild. Results showed that the 31–40 age group, healthcare providers other than nurses and physicians, Saudi nationality, and healthcare providers with sleeping disorders before COVID-19 were associated with anxiety levels. Several factors might contribute to higher anxiety levels among Saudi healthcare providers: such as marital status, socioeconomic status, having a chronic health condition, fear of contracting the virus, or living with a person at high risk for severe illness. To further understand the level of anxiety among healthcare providers during the COVID-19 pandemic in Saudi Arabia, longitudinal and mixed-method research is needed. When these factors were examined simultaneously, study found that gender, nationality, and sleeping disorder before Covid-19 are the main significant factors for anxiety levels. It was found that being a female, Saudi nationality, and having sleeping disorder before Covid-19 greatly increase the odds of having higher level of anxiety.

### Consent for publication

Participation in the study was voluntary and participants were ensured that information gathered for the study would be kept confidential and will be used for the study purposes only.

## LIST OF ABBREVIATIONS

| | |
|---|---|
| **COVID-19** | Coronavirus Disease 2019 |
| **SARS-CoV-2** | Severe acute respiratory syndrome coronavirus 2 |
| **GAD** | Generalized anxiety disorder |
| **WHO** | World Health Organization |
| **IRB** | Institutional Review Board |
| **HCWs** | Health care works |
| **SPSS** | Statistical Package for the Social Sciences |
| **SD** | Standard deviation |
| **IQR** | Interquartile range |
| **SDS** | Self-Rating Depression Scale |

## ACKNOWLEDGEMENTS

The authors declare no conflict of interest in preparing this article, authors thank the referee for constructive comments.

### Funding
The authors received no funding for this work.

### Competing Interests
Abbas Al Mutair and Abbas Shamsan were employed by the Dr. Sulaiman Al Habib Medical Group. The other authors declare that they have no competing interests.

### Author Contributions
- Abbas Al Mutair conceived and designed the experiments, performed the experiments, authored or reviewed drafts of the paper, and approved the final draft.
- Alya Al Mutairi conceived and designed the experiments, analyzed the data, prepared figures and/or tables, and approved the final draft.
- Yasmine Alabbasi conceived and designed the experiments, authored or reviewed drafts of the paper, and approved the final draft.
- Abbas Shamsan conceived and designed the experiments, authored or reviewed drafts of the paper, and approved the final draft.
- Sana Al-Mahmoud conceived and designed the experiments, prepared figures and/or tables, authored or reviewed drafts of the paper, and approved the final draft.
- Saad Alhumaid performed the experiments, authored or reviewed drafts of the paper, and approved the final draft.
- Muhammad Zeshan Arshad analyzed the data, prepared figures and/or tables, and approved the final draft.
- Mansour Awad performed the experiments, prepared figures and/or tables, and approved the final draft.
- Ali Rabaan conceived and designed the experiments, prepared figures and/or tables, and approved the final draft.

### Ethics
The following information was supplied relating to ethical approvals (*i.e.*, approving body and any reference numbers):

The Institutional Review Board at Dr. Sulaiman Al Habib Medical Group approved this research (IRB Log No. RC20.06.88-03).

### Data Availability
Raw data are available as a Supplementary File.

### Supplemental Information
Supplemental information for this article can be found online at http://dx.doi.org/10.7717/peerj.12119#supplemental-information.

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
