# Peer review of "Level of anxiety among healthcare providers during COVID-19 pandemic in Saudi Arabia: cross-sectional study"

_PeerJ, doi:10.7717/peerj.12119_

## Round 0.1 · original submission · Major Revisions

Please remember that It is relevant to give sufficient details of the data analysis process, including at least, the tests used. Also, make sure to include more information about how data collection and recruitment were done. A very relevant question is to indicate if the questionnaire was validated, if so, please indicate the reference.

Authors are not encouraged to include all citations suggested by reviewers. Only if authors consider that those citations are really necessary or relevant, you can include them. My final decision will not be based on suggested citations but on the quality of the work.

Reviewer 1 ·

Basic reporting

Manuscript with a relevant theme in the current context.
Methods:
-The authors do not detail the data analysis process, such as the tests used.
More detail is needed on how data collection and recruitment was done.
-Was the questionnaire validated?
Results
- It is suggested to divide the characteristics of the results by the outcome and present relevant statistics.
-Present the study flowchart.
-Review the discussion and present strengths.
-The conclusion is beyond the objectives.

Experimental design

No comment.

Validity of the findings

Statistical analysis data do not support discussion.

Additional comments

Manuscript with a relevant theme in the current context.
Methods:
-The authors do not detail the data analysis process, such as the tests used.
More detail is needed on how data collection and recruitment was done.
-Was the questionnaire validated?
Results
- It is suggested to divide the characteristics of the results by the outcome and present relevant statistics.
-Present the study flowchart.
-Review the discussion and present strengths.
-The conclusion is beyond the objectives.

Reviewer 2 ·

Basic reporting

References are insufficient in introduction and methods.
Cite the following: Magnavita N, Chirico F, Garbarino S, Bragazzi NL, Santacroce E, Zaffina S. SARS/MERS/SARS-CoV-2 Outbreaks and Burnout Syndrome among Healthcare Workers. An umbrella Systematic Review. Int J Environ Res Public Health. 2021;18(8):4361. Doi: 10.3390/ijerph18084361.Chirico F, Magnavita N. The Crucial Role of Occupational Health Surveillance for Health-care Workers During the COVID-19 Pandemic. Workplace Health & Safety. 2021;69(1):5-6. doi:10.1177/2165079920950161. Chirico F, Ferrari G, Nucera G, Szarpak L, Crescenzo P, Ilesanmi O. Prevalence of anxiety, depression, burnout syndrome,and mental health disorders among healthcare
workers during the COVID-19 pandemic: A rapid umbrella review of systematic reviews. J Health Soc Sci. 2021;6(2).doi: 0.19204/2021/prvl7.Chirico F, Nucera G. Tribute to healthcare operators threatened by COVID-19 pandemic. J Health Soc Sci. 2020;5(2):165-168. 10.19204/2020/trbt1, Chirico F, Nucera G, Magnavita N. Protecting the mental health of healthcare workers during the COVID-19 emergency. BJ Psych International. 2020. 1-6. Doi: 10.1192/bji.2020.39

English should be largely improved

Experimental design

Research question should be reformulated
The association between socio-demographic characteristics and levels of anxiety and depression should be defined in terms of predictors, by using ORs.
In methods the authors do not report important ethical aspects and do not describe how they use dependent variable.
I suggest using multivariate regression logistics where dependent variable (yes/no) is anxiety or depression and individual and occupational predictors are independent variables

Validity of the findings

Conclusions are supported by results

---

## Round 0.2 · accepted · Accept

Authors have revised the manuscript according to reviewers comments.

Reviewer 1 ·

Basic reporting

The changes were done.

Experimental design

The changes were done.

Validity of the findings

The manuscript improved.

Reviewer 2 ·

Basic reporting

The reporting is clear and unambiguous. Literature references are sufficient.

Experimental design

Research is within aims and scope of the journal.
Research question is well defined, relevant and meaningful.
Methods are described sufficiently

Validity of the findings

All data look robust and conclusions are well stated